# SELF-GUIDED MASKED AUTOENCODERS FOR DOMAIN-AGNOSTIC SELF-SUPERVISED LEARNING

**Johnathan Xie** [*]
Stanford University

**Yoonho Lee**
Stanford University

**Annie S. Chen**
Stanford University

**Chelsea Finn**
Stanford University

## ABSTRACT

Self-supervised learning excels in learning representations from large amounts of unlabeled data, demonstrating success across multiple data modalities. Yet, extending self-supervised learning to new modalities is non-trivial because the specifics of existing methods are tailored to each domain, such as domain-specific augmentations which reflect the invariances in the target task. While masked modeling is promising as a domain-agnostic framework for self-supervised learning because it does not rely on input augmentations, its mask sampling procedure remains domain-specific. We present Self-guided Masked Autoencoders (SMA), a fully domain-agnostic masked modeling method. SMA trains an attention based model using a masked modeling objective while learning masks to sample without any domain-specific assumptions. We evaluate SMA on three self-supervised learning benchmarks in protein biology, chemical property prediction, and particle physics. We find SMA is capable of learning representations without domain-specific knowledge and achieves state-of-the-art performance on these three benchmarks. [1]

## 1 INTRODUCTION

Self-supervised learning is a powerful framework for learning representations and is appealing especially due to its ability to leverage large amounts of unlabeled data. The pursuit of scaling self-supervised learning to larger datasets has led to impressive results in the natural language processing (Devlin et al., 2018; Brown et al., 2020; Raffel et al., 2020) and computer vision (Chen et al., 2020; He et al., 2022; Oquab et al., 2023) domains. While many works in machine learning have aimed to improve results by introducing domain-specific inductive biases, various studies have observed that the advantages of inductive biases diminish at large data scales (Dosovitskiy et al., 2020; Geirhos et al., 2018; Sutton, 2019). Therefore, it is striking that many self-supervised learning methods rely closely on domain-specific knowledge (Chen et al., 2020; Grill et al., 2020; Caron et al., 2021). Simultaneously, the domain-specific choices make it difficult to apply these self-supervised learning methods to new domains. Motivated by the potential for improved performance and greater applicability to new domains, we aim to develop a domain-agnostic self-supervised learning algorithm.

One common class of self-supervised learning approaches is contrastive learning (Chen et al., 2020; Tian et al., 2020; Oord et al., 2018; Wu et al., 2020) which operates by encouraging models to represent similar examples in nearby parts of the representation space. Despite the simple core idea, these methods necessarily rely on domain-specific augmentations to generate examples with similar semantic content (Chen et al., 2020; Oord et al., 2018; Grill et al., 2020). An alternative approach to self-supervised learning is masked modeling (Devlin et al., 2018; He et al., 2022) where the model is trained to reconstruct parts of the input that are masked out. Towards the goal of a domain-agnostic self-supervised learning method, masked modeling is promising because the idea of masking out parts of an input is widely applicable and various works show strong performance across a diverse array of data modalities (Devlin et al., 2018; He et al., 2022; Xie et al., 2022; Baevski et al., 2022; 2020; Yu et al., 2022; Wang et al., 2022). However, generalizing these methods across different domains remains challenging because data modalities differ in how raw input values are related. Previous methods have used domain-specific mask sampling methods or different tokenizers for each domain.

---

[*]Correspondence to jwxie@stanford.edu
[1]We make the code available at this link.

While these strategies have been effective, they make masked modeling non-trivial to extend to new domains because they require the development of new tokenizers which are also error prone, lack robustness, and are difficult to maintain (Xue et al., 2022; Jaegle et al., 2021b). Rather than rely on tokenizers or domain-specific assumptions, we demonstrate that relations between raw inputs can be learned during masked pre-training which in turn simultaneously drives the sampling of useful masks for masked modeling.

The main contribution of this paper is Self-guided Masked Autoencoders (SMA), a self-supervised learning method based on masked modeling that is entirely domain-agnostic as it does not use any form of tokenizer or priors regarding the structure of raw inputs. For both cross-attention (Lee et al., 2019; Jaegle et al., 2021a) and self-attention (Vaswani et al., 2017) architectures, we compute masks based on the attention map of the first encoding layer during masked prediction training. This way of producing input masks is motivated by recent observations that such attention maps correspond to semantic regions of the input (Jaegle et al., 2021a; Caron et al., 2021). We perform experiments on three data modalities: protein biology, chemical property prediction, and particle physics, and on all three domains, SMA achieves state-of-the-art performance while remaining domain-agnostic. Our results demonstrate that much of the domain-specific knowledge used by prior methods to guide model training can be found within the unlabeled data.

## 2 RELATED WORK

**Masked modeling.** A common method for self-supervised learning is for a model to receive to receive a partially masked input sequence and learn to predict the masked values. Prior works have used this approach in the language domain (Devlin et al., 2018; Liu et al., 2019; Raffel et al., 2020; Lewis et al., 2019) and shown that it scales well to large datasets and models (Chowdhery et al., 2022; Brown et al., 2020). Recent works extend masked modeling to the image (He et al., 2022; Xie et al., 2022), video (Feichtenhofer et al., 2022), audio (Schneider et al., 2019; Baevski et al., 2020; Xu et al., 2022), and point cloud (Yu et al., 2022; Pang et al., 2022) domains. A key advantage of masked modeling approaches is that they are substantially less reliant on domain-specific augmentations (He et al., 2022) compared to joint-embedding approaches (Chen et al., 2020; Oord et al., 2018; Grill et al., 2020) and are more stable as they are not susceptible to representation collapse. However, masked modeling methods still require tailored domain-specific masks or tokenizers to learn useful representations (Xie et al., 2022; Raffel et al., 2020). In comparison to prior works, we propose a system to generate a challenging masking objective based on a network's attentions, alleviating the need for domain-specific masks and tokenizers.

**Domain-agnostic learning.** With the introduction of the Transformer architecture (Vaswani et al., 2017), which can effectively process sequences of data vectors with few assumptions, representation learning methods have become more unified across domains. Though the Transformer has become ubiquitous for processing high-level input tokens, the preprocessing required for each domain still differs drastically and effective tokenization methods are non-trivial to develop for new domains. Recently, new cross-attention-based architectures (Jaegle et al., 2021b;a; Lee et al., 2019; Yu et al., 2023) have enabled processing of raw inputs such as pixels or characters. These architectures demonstrate a path to truly domain-agnostic learning with a model that can be applied out of the box to any domain. Our method leverages these new architecture designs to create end-to-end domain-agnostic self-supervised learning without the need for tokenization.

**Learning corruptions.** Related to our work are methods for automatically learning augmentations and corruptions for representation learning. Early methods simply involved determining which hand-designed augmentations to apply to a training example (Cubuk et al., 2018; Lim et al., 2019; Cubuk et al., 2020; Suzuki, 2022), however recent approaches (Suzuki, 2022; Tamkin et al., 2020) have leveraged fewer inductive biases with regards to the domain by automating a majority of the augmentation pipeline. Most related to our work are methods which learn to sample masks in contrastive learning (Shi et al., 2022) and video pretraining (Bandara et al., 2023). Compared to prior works, ours does not rely on the use of any domain-specific tokenization or additional augmentations to learn representations.

**Deep learning for science.** Recently there have been many advances in applying deep learning techniques in various scientific domains. These methods often leverage scientific knowledge of each domain to design architectures (Brandes et al., 2022; Zhou et al., 2023) or representation learning

methods (Zhou et al., 2023; Ahmad et al., 2022). However, these methods are reliant on a few known invariances (Zhou et al., 2023; Qu and Gouskos, 2020) within each scientific domain or additional engineered features (Brandes et al., 2022) to improve training which is challenging to extend to new domains. In comparison to these works, we propose a framework to learn representations from scratch in a data-driven manner without reliance on domain knowledge leading to more generally applicable representation learning. We demonstrate that a strong masking policy that corresponds to contiguous parts can be learned without any inductive priors or tokenization. Our modeling method demonstrates a path toward representation learning that is based solely on provided data and is not reliant on any prior knowledge (LeCun et al., 2015).

## 3 BACKGROUND

In this section, we provide an overview of the masked prediction objective which we build on top of in our method, attention-based modeling, and we formally define the concept of domain-agnostic learning.

### 3.1 MASKED PREDICTION

Our method is based on masked prediction, a commonly used objective for self-supervised learning. Given only a subset of the input tokens, and the remaining tokens masked, the model is trained to predict these masked tokens. We consider an input sequence consisting of $n$ tokens $X = (c_1, \ldots, c_n)$. We denote the masked and unmasked indices as $M \subset \{1, \ldots, n\}$ and $U = \{1, \ldots, n\} - M$, with a total of $m$ masked tokens and $u = n - m$ unmasked tokens. The task of predicting the masked tokens $\{c_i\}_{i \in M}$ given the unmasked tokens $\{c_i\}_{i \in U}$ is most informative when the model must rely on the semantic context given by the unmasked tokens to make accurate predictions. Because many modalities such as image or video data have near-duplication or strong dependencies in their low-level inputs a model can simply learn to predict masked tokens by interpolating the value of the nearest unmasked tokens, which is not very helpful for representation learning.

Thus, to ensure that the model learns to infer semantic context from the unmasked tokens, the masking procedure must be carefully designed to ensure that the masked prediction problem is not trivial. For example, in the language domain, the masking procedure typically masks out a contiguous span of tokens such as a word (Devlin et al., 2018; Liu et al., 2019; Raffel et al., 2020). In the image domain, the mask typically corresponds to a contiguous region of pixels (He et al., 2022; Xie et al., 2022). As we are interested in developing a fully domain-agnostic method, we refrain from assuming any prior knowledge about the structure of low-level inputs. Instead, we propose a data-driven approach to producing masks that correspond to contiguous parts.

### 3.2 DOMAIN-AGNOSTIC LEARNING

The Perceiver (Jaegle et al., 2021b) architecture introduces a notion of a domain-agnostic process by arguing that a full shuffling of input tokens along with their positional embeddings should produce an equivalent output. We extend this definition to a full learning process in order to formalize the definition of domain-agnostic that we adhere to throughout our paper. Formally, for an unlabeled training set of $k$ raw data samples, $\{X^{(1)}, \ldots, X^{(k)}\}$, where each $X^{(i)} \in \mathbb{R}^{n \times d_{raw}}$, let $\mathcal{P}$ represent the set of all permutations of indices $\{1, \ldots, n\}$. A learning process $\mathcal{L}$ for a network $\mathcal{N}$ can be defined as domain-agnostic if

$$\forall p \in \mathcal{P}, \mathcal{L}(\mathcal{N}, \{X^{(1)}, \ldots, X^{(k)}\}) = \mathcal{L}(\mathcal{N}, \{pX^{(1)}, \ldots, pX^{(k)}\}) \qquad (1)$$

The process of reshaping the raw modality inputs into the shape $n \times d_{raw}$ must only involve knowledge of what dimensionality the data is and is generally just a simple flattening operation with the possibility of padding if necessary.

This definition of domain-agnostic differs from the definition of permutation invariance in that it requires a strictly *consistent* shuffling of the individual tokens within each data sample $X^{(i)}$ across the dataset. This means that positional information can still be learned by the network, but there is no prior knowledge of how related two input tokens are or their causality.

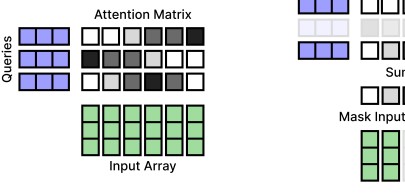 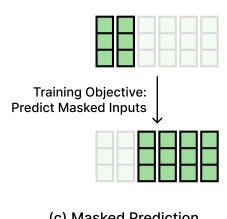

| (a) Compute Attention Map | (b) Randomly Select Queries | (c) Masked Prediction |

Figure 1: **Self-guided Masked Autoencoders (SMA).** (a) We first extract the cross-attention map from latent queries onto the full input array. (b) Next, add the rows of the attention matrix corresponding to a randomly selected subset of the queries, and produce an input mask by selecting the largest values of the attention sum. (c) Our pre-training objective is to reconstruct the original inputs from the masked input sequence.

### 3.3 ATTENTION

To compute masks in a completely domain-agnostic fashion, we leverage information pertaining to semantic groups encoded in attention maps (Jaegle et al., 2021b; Caron et al., 2021). Two common methods for computing attention across an input are self-attention (Vaswani et al., 2017) where query and key values are projected from the same set of vectors and cross-attention (Jaegle et al., 2021b; Lee et al., 2019) where query values are projected from a separate set of vectors. To compute attention maps, first we must embed the raw inputs $X \in \mathbb{R}^{n \times d_{raw}}$ by using a linear projection $E \in \mathbb{R}^{d_{raw} \times d_{embed}}$ and positional embeddings learned from scratch $P \in \mathbb{R}^{n \times d_{embed}}$ to produce $XE + P = \hat{X} \in \mathbb{R}^{n \times d_{embed}}$. From these inputs, we produce attention keys $K_e \in \mathbb{R}^{n \times d_k}$ using a linear projection. The queries for attention $Q_e \in \mathbb{R}^{l \times d_k}$ [2] can either be projected from the embedded inputs $\hat{X}$ in the case of self-attention, or learnable latents in the case of cross-attention. From these queries and keys, we can compute the initial unnormalized attention map:

$$A = \frac{Q_e K_e^\top}{\sqrt{d_k}} \in \mathbb{R}^{l \times n}. \tag{2}$$

As our method is applicable to both self-attention and cross-attention maps, we do not specifically denote the difference between these two cases in the rest of the paper.

## 4 SELF-GUIDED MASKED AUTOENCODERS

In this section, we describe self-guided masked autoencoders (SMA), a domain-agnostic approach to self-supervised learning that samples masks from the masked prediction model's own attention representations. Our core mask extraction procedure is visualized in Figure 1, and we provide an overview of the full training loop in Algorithm 1.

Intuitively, masking contiguous groups of highly correlated tokens is desirable for masked modeling approaches because it requires more complex modeling of the input, which in turn facilitates the learning of representations that

---

**Algorithm 1** SMA training

**Input:** Data $D = \{X^{(1)}, \ldots\}$, reconstruction loss $\mathcal{L}$, initial parameters $\theta_0$, masking ratio $r$.

**while** Not converged **do**
    $X \sim D$
    $A \leftarrow Attention(\theta; X)$          ▷ Sec. 3.3
    $\hat{A} \leftarrow ApplyMask(A, r)$          ▷ (5)
    $H \leftarrow Encode(\theta; \hat{A}, X)$       ▷ Sec. 4.2
    $O \leftarrow Upsample(\theta; H)$         ▷ (6)
    $\theta \leftarrow \theta - \alpha \nabla_\theta \ell(O, X)$     ▷ Sec. 4.2
Return $\theta$

---

capture important features. Thus, our goal is to identify and mask tokens that are highly correlated in a single group. One potential way to achieve this is by leveraging the masked prediction model's own internal attention representations, which have been shown to correspond to semantic regions of the input (Jaegle et al., 2021b; Caron et al., 2021). By using these attention representations to guide the masking process, we can mask highly correlated and redundant tokens. This is especially important in the scientific domains we explore, where identifying all correlated parts of the input would otherwise require specialized knowledge of the scientific domain.

---

[2]In the case of self-attention, $l = n$.

## 4.1 SELECTING TOKENS TO MASK

We now describe our method for selecting tokens for masking from an attention map (2) of either self-attention (Vaswani et al., 2017) or cross-attention (Lee et al., 2019; Jaegle et al., 2021b). We find in practice that each query assigns large attention to highly correlated groups of inputs. Therefore, to mask all the inputs of a single "group," we can mask the top attentions of a given query. However, the naive application of this masking algorithm is problematic for a few reasons. First, because the top attention tokens chosen for each query have significant overlap, we often do not actually mask the target masking ratio, and this issue worsens as $n$ increases. Second, the iterative nature of the algorithm leads to poor complexity with the repeated top-k operations, and poor parallelization.

Instead, we can compute an approximation of this procedure that we find to be equally effective in practice while achieving the desired masking ratio and parallelizing well.

To select masked indices, we first randomly choose a subset of query indices $\mathcal{R} \subset \{1, \ldots, l\}$. Then, with a masking ratio of $r$ the function to generate a mask of $nr$ elements for a given attention map $A$ is defined as

$$KeepTopK(v, k)_i = \begin{cases} 0 & \text{if } v_i \text{ is in the top k elements of v} \\ -\infty & otherwise. \end{cases} \tag{3}$$

$$AttentionMask(A, k) = KeepTopK\left(\left\langle \sum_{i \in R} A_{0i}, \sum_{i \in R} A_{1i}, \ldots \sum_{i \in R} A_{ni} \right\rangle, k\right) \tag{4}$$

We follow Shazeer et al. (2017) in defining the $KeepTopK$ operation. In practice, attention architectures often use a multi-headed (Vaswani et al., 2017) approach to capture different relations in a single attention layer. Therefore, rather than an attention matrix $A \in \mathbb{R}^{l \times n}$, the attention matrix is often of shape $A \in \mathbb{R}^{h \times l \times n}$. To select queries to be masked across heads, we find we can aggregate the attentions after softmax normalization, but prior to the masking operation by summing across the head dimension. This aggregation operation is performed only for the purpose of computing a mask, and does not affect the final attention mask. Then, we can mask attentions across heads such that there is no information flow from masked tokens. With this, the resulting masked attention matrix after masking can be written as

$$\hat{A} = SoftMax(AttentionMask(\sum_{i=0}^{h} SoftMax(A_i), \lfloor nr \rceil) + A).^{3\ 4} \tag{5}$$

This full operation can be viewed as an approximation of selecting the top attentions of a subset of the queries then masking them by setting their values to a large negative value prior to the softmax operation. Our masking design is especially efficient and in practice does not add any significant compute as it only requires an extra summation, top-k, and softmax operation. Additionally, the entire masking operation is contained to only the first layer as the masked inputs have no attention value and are dropped in the layer output preventing any information flow to the model (He et al., 2022). The full masking operation is visually summarized in parts (a) and (b) of Figure 1.

## 4.2 MASKED RECONSTRUCTION TRAINING

Following the initial attention encoding, these hidden vectors are further processed using repeated Transformer (Vaswani et al., 2017) layers. To create reconstruction predictions, we follow a process similar to prior works (He et al., 2022; Jaegle et al., 2021a). Specifically, we upsample the processed hidden vectors $H \in \mathbb{R}^{l \times d_v}$ to the same dimensionality as the input by projecting decoding query values $Q_o \in \mathbb{R}^{n \times d_k}$ from $P$, the model's positional embeddings. Additionally, we project keys $K_o \in \mathbb{R}^{l \times d_k}$ and values $V_o \in \mathbb{R}^{l \times d_v}$ from $H$. Using these values, we compute the initial output:

$$O = SoftMax(\frac{Q_o K_o^\top}{\sqrt{d_k}})V_o \in \mathbb{R}^{n \times d_v} \tag{6}$$

---

[3] The one-dimensional masking vector is expanded in the two initial dimensions, h times then l times in order to fully mask across these dimensions.

[4] $\lfloor nr \rceil$ means integer rounding.

| Method | Domain Agnostic | Remote Homology | Fluorescence | Stability |
|---|:---:|:---:|:---:|:---:|
| ProteinBERT (Brandes et al., 2022) | ✗ | 0.22 | 0.66 | 0.76 |
| No Pretrain | | 0.10 | 0.60 | 0.66 |
| Random Masking | ✓ | 0.22 | 0.67 | 0.76 |
| SMA (ours) | | **0.23** | **0.68** | **0.80** |

Table 1: **Transfer learning comparison on TAPE Benchmark (Rao et al., 2019) downstream tasks.** We report accuracy for Remote Homology and Spearman correlation for Fluorescence and Stability.

Then, we compute raw output predictions by using an additional MLP to project $O$ to the original raw value dimension $d_{raw}$. Finally, we compute a reconstruction loss using either mean-squared error for continuous raw values or cross-entropy for discrete raw values. This reconstruction loss is computed only on masked indices $m$ as shown in part (c) of Figure 1. The masked prediction model parameters are updated via mini-batch gradient descent to minimize the reconstruction loss. Importantly, only a single masked prediction model is used during the entire training process, as our method learns to sample masks while simultaneously optimizing the masked prediction objective.

## 5 EXPERIMENTS

In this section, we aim to answer the following experimental questions:

1. On a wide range of domains, can SMA learn masks that are effective for learning representations in masked modeling *without any domain knowledge*?

2. Can SMA outperform prior domain-specific methods on representation learning benchmarks across a wide range of domains using both self-attention and cross-attention architectures?

To test the efficacy of our method we study pre-training over an unlabeled training set followed by downstream transfer to some related task within the same domain. We compare against the prior state-of-the-art in the protein biology, chemistry, and particle physics domains. Values in the protein biology and particle physics domains are the mean of three random seed values, while in the chemistry domain we take the mean of values across three scaffold splits commonly used by prior works (Ramsundar et al., 2019). For all experiments, we provide further details on hyperparameters and model configurations in Section B of the appendix.

**Model Architecture.** It is important that our method is applicable to both self-attention and cross-attention architectures since often the raw form of many modalities have very long sequence lengths causing standard self-attention to be too costly to compute over such long sequences. Therefore, in the protein domain we adopt a cross attention architecture based on the Perceiver IO (Jaegle et al., 2021a) model. However, the max sequence length in the chemistry and particle physics domains are relatively short, 256 and 28 respectively. Therefore, we use a standard transformer self-attention architecture (Vaswani et al., 2017) in these domains. In order to vectorize inputs for discrete modalities such as text, protein, and molecule sequences we map each character to a learned vector which is then summed with a learned positional embedding. In continuous domains such as images and particle physics, we use a linear projection for each continuous input to project them to the same dimension as positional embeddings which are then summed.

### 5.1 PROTEIN PROPERTY PREDICTION

Unlike well studied discrete domains such as natural language, the protein biology domain does not have well-established effective inductive biases or tokenizers. We apply our method in this domain and find that it is capable of learning a non-trivial masking strategy and outperforms hand designed biases. We follow the benchmark setting of TAPE (Rao et al., 2019; Tamkin et al., 2021) which involves pre-training over the Pfam training set for 10 epochs. To assess self-supervised representation quality, we fine-tune over two regression tasks (Fluorescence and Stability) where we report Spearman correlation and a classification task (Remote Homology) where we report accuracy.

We find that SMA is able to outperform both the previous state-of-the-art methods and the random masking baseline. Compared to the prior state-of-the-art work of ProteinBERT (Brandes et al., 2022)

| | Domain Agnostic | ROC (higher is better ↑) | | | RMSE (lower is better ↓) | | |
|---|---|---|---|---|---|---|---|
| | | BBBP | BACE | HIV | ESOL | FreeSolv | Lipo |
| **ChemBERTa-2** (Ahmad et al., 2022) | | | | | | | |
| MLM-5M | | 0.701 | 0.793 | - | - | - | 0.986 |
| MLM-10M | | 0.696 | 0.729 | - | - | - | 1.009 |
| MLM-77M | ✗ | 0.698 | 0.735 | - | - | - | 0.987 |
| MTR-5M | | 0.742 | 0.734 | - | - | - | 0.758 |
| MTR-10M | | 0.733 | 0.783 | - | - | - | 0.744 |
| MTR-77M | | 0.728 | 0.799 | - | - | - | 0.798 |
| Uni-Mol-20M (Zhou et al., 2023) | ✗ | 0.729 | **0.857** | **0.808** | 0.788 | 1.48 | **0.603** |
| No Pretrain | | 0.694 | 0.763 | 0.730 | 0.877 | 1.58 | 0.962 |
| Random Masking-1.6M | ✓ | 0.723 | 0.798 | 0.765 | 0.642 | 1.22 | 0.631 |
| SMA-1.6M (Ours) | | **0.750** | 0.843 | 0.789 | **0.623** | **1.09** | **0.609** |

Table 2: MoleculeNet property prediction comparison using deepchem Ramsundar et al. (2019) scaffold splits. We compare against the state-of-the-art SMILES-based model, ChemBERTa-2 Ahmad et al. (2022), and the state-of-the-art graph-based model Uni-Mol Zhou et al. (2023).

| Architecture | Pretraining | Accuracy (%) | | |
|---|---|---|---|---|
| | | 1k | 10k | 100k |
| TabNet (Arik and Pfister, 2021) | None | 57.47 | 66.66 | 72.92 |
| TabNet (Arik and Pfister, 2021) | Learned Masking | 61.37 | 68.06 | 73.19 |
| Transformer (Vaswani et al., 2017) | None | 65.27 | 70.86 | 74.05 |
| Transformer (Vaswani et al., 2017) | Random Masking | 68.07 | 72.57 | 76.54 |
| Transformer (Vaswani et al., 2017) | Guided Masking (Ours) | **69.47** | **74.04** | **77.88** |

(a) **Transfer learning comparison on HIGGS benchmark** test set. We finetune pretrained models over training subsets of different sizes to assess self-supervised learning quality.

| Method | Accuracy (%) |
|---|---|
| TabNet (Arik and Pfister, 2021) | 71.9 |
| SNN (Klambauer et al., 2017) | 72.2 |
| AutoInt (Song et al., 2019) | 72.5 |
| GrowNet (Badirli et al., 2020) | 72.2 |
| MLP (Gorishniy et al., 2021) | 72.3 |
| DCN2 (Wang et al., 2021) | 72.3 |
| NODE (Popov et al., 2019) | 72.6 |
| ResNet (Gorishniy et al., 2021) | 72.7 |
| CatBoost (Prokhorenkova et al., 2018) | 72.6 |
| XGBoost (Chen and Guestrin, 2016) | 72.7 |
| FT-T (Gorishniy et al., 2021) | 72.9 |
| Transformer-PLR (Gorishniy et al., 2022) | 73.0 |
| Transformer Scratch | 72.7 |
| Transformer Random Masking | 74.0 |
| Transformer Guided Masking (Ours) | **74.8** |

(b) **HIGGS small benchmark classification accuracy**. Random masking and Guided masking methods are pretrained over only the HIGGS small training set.

which uses domain knowledge to design a custom architecture and additional sequence labels in their supervisory task, our method improves self-supervision via more challenging masks.

## 5.2 CHEMICAL PROPERTY PREDICTION

We compare SMA to the previous state-of-the-art works on MoleculeNet regression and classification tasks (Wu et al., 2018) which span a wide range of chemical property prediction tasks. Prior works are mainly either graph-based approaches which represent molecules as 3D graphs, or SMILES (Weininger, 1988) strings. We choose to represent input molecules as SMILES strings which allows us to apply our method with minimal modification. Whereas other SMILES-based methods use chemistry specific tokenizers that aggregate molecule information, for the sake of generality we choose to map strings to discrete values using a simple UTF-8 character mapping. Following prior work, we use the cross validation splits from deepchem (Ramsundar et al., 2019) which uses scaffold splitting to test generalization across different molecule types. We compare against the state-of-the-art SMILES-based model, ChemBERTa-2, and the state-of-the-art graph-based model, Uni-Mol. We pre-train on the training set of the guacamol (Brown et al., 2019) dataset containing 1.6M molecules, an order of magnitude smaller than the training sets of both ChemBERTa-2 (Ahmad et al., 2022) and Uni-Mol (Zhou et al., 2023). Our method outperforms all 6 ChemBERTa-2 configurations on the three shared benchmark tasks and outperforms Uni-Mol on three out of six benchmark tasks while being comparable in a fourth. Compared to Uni-Mol which uses atom location information and augments their data by generating atom conformations, our method is far simpler and is a direct generalization of our algorithm to this discrete domain.

## 5.3 PARTICLE PHYSICS PROCESS CLASSIFICATION

We use the HIGGS (Whiteson, 2014) particle physics dataset where the task is to distinguish between a Higgs Boson process and a background process. The input consists of 21 kinematic values and 7 high-level features calculated from the kinematic features. We benchmark under two different settings. First, we compare using the same self-supervised learning setting as TabNet (Arik and Pfister, 2021)

| Pretraining Method | Domain Agnostic | MNLI-(m/mm) 392k | QQP 363k | QNLI 108k | SST-2 67k | CoLA 8.5k | STS-B 5.7k | MRPC 3.5k | RTE 2.5k | Avg |
|---|---|---|---|---|---|---|---|---|---|---|
| Word | ✗ | **79.0** | 88.3 | **87.4** | **89.9** | 40.8 | 82.2 | **82.3** | 54.5 | 75.6 |
| None | | 43.5 | 75.4 | 61.6 | 60.3 | 0.0 | 28.0 | 70.3 | 49.1 | 48.5 |
| Random | ✓ | 75.9 | 88.7 | 84.7 | 86.8 | 36.2 | 83.4 | 76.2 | 55.2 | 73.4 |
| SMA (ours) | | 77.5 | **89.1** | **87.4** | 88.3 | **43.3** | **84.6** | 80.9 | **62.1** | **76.7** |

Table 4: **Transfer learning comparison on GLUE benchmark** evaluation set. Tasks are ordered in decreasing order of training set size. We report F1 score for QQP and MRPC, Spearman's $\rho$ for STS-B, and accuracy for other tasks. We find that SMA outperforms the three baselines (No Pretrain, Random, and Word) with the same architecture, and at the current scale, we find that our method can outperform the popular word-masking method by approximately 1.5% when averaged across GLUE tasks.

where we pre-train over the full 10.5m sample training set, then fine-tune over a selected subset of the training set of size 100,000, 10,000, and 1000 samples and report binary classification accuracy in Table 3a. Second, we consider a supervised tabular dataset split consisting of 62,752 training samples and 19,610 test samples (Gorishniy et al., 2021). Under this setting we pre-train only over the same labeled set available to all supervised methods then fine-tune of the labeled set and report results in Table 3b. Because HIGGS is a common tabular data benchmark, all prior methods are technically domain-agnostic as there does not exist general domain-specific knowledge pertaining to tabular datasets.

We find that under the self-supervised learning setting in Table 3a, SMA is able to provide a consistent improvement in classification accuracy over both no pre-training and random masked pre-training across all fine-tuning dataset sizes. Additionally, we find our pre-training method to yield substantially greater gains over supervised models when compared to the learned masking method of TabNet (Arik and Pfister, 2021) which aims to mask out irrelevant features. Under the supervised tabular split in Table 3b, where the pre-training set and fine-tuning labeled sets are identical we still find SMA yields a substantial improvement. Whereas all prior supervised methods differ by only 0.6% accuracy, SMA is able to outperform the previous state-of-the-art by 1.9% demonstrating our pre-training method can yield significant gains unmatched by using only supervised training.

## 5.4 COMPARISON WITH DOMAIN-SPECIFIC MASKS

While our method is best used for domains without strong priors such as the aforementioned scientific domains, we aim to compare the viability of generated masks for representation learning with hand-designed masks from well studied domains such as the image and language domains. Additionally, we qualitatively assess generated masks which are included in Section C of the appendix. Due to these two domains having long sequence lengths in their raw form, we use a cross attention architecture for both domains.

### 5.4.1 NATURAL LANGUAGE TASKS

For natural language pre-training, we use a concatenation of English Wikipedia and BooksCorpus and assess the representation quality by fine-tuning over the GLUE (Wang et al., 2018) benchmark excluding the WNLI dataset as is common practice since BERT (Devlin et al., 2018). We average these scores over three fine-tuning seeds. To map text sequences to discrete values, we use a basic UTF-8 character level mapping (Jaegle et al., 2021a).

To assess representation quality, we evaluate on the development set of the GLUE dataset and report the unweighted average of F1 scores for QQP and MRPC, spearman correlations for STS-B, and accuracy for other tasks in Table 4. We use a single set of hyperparameters across all tasks which is further described in the appendix along with additional details on our model architecture and pre-training hyperparameters

We compare our method to three alternatives using the same architecture: (1) No pre-training, (2) Random masking, (3) Word-level masking, a domain-specific masking method. For word masking, we reproduce the results of Perceiver IO (Jaegle et al., 2021a) [5] where a word is considered a sequence

---

[5] Due to computational constraints our model and pre-training set size are considerably smaller than the original paper.

| Pretraining Method | Domain Agnostic | w/o augmentation | w/ augmentation |
|---|---|---|---|
| Patch | ✗ | **79.6** | 84.3 |
| None | | 52.5 | 62.7 |
| Random | ✓ | 77.6 | 85.6 |
| SMA (ours) | | **78.5** | **86.1** |

Table 5: **Top-1 accuracy transfer learning comparison on ImageNet-100** evaluation set. We consider a setting of both using augmentation and not using augmentation during downstream finetuning. All models are first pretrained for 200 epochs over the training set using different masking methods. We find that our method outperforms random masking both with and without augmentation during downstream classification.

of space delimited characters. SMA outperforms the three baselines with the same architecture, and at the current scale, we find that our method can outperform the word-masking method by approximately 1.5% when averaged across GLUE tasks.

### 5.4.2 IMAGE CLASSIFICATION

To assess image domain self-supervised learning, we pre-train all models on the ImageNet-100 (Tian et al., 2020) subset. In all pre-training runs we also resize images to a resolution of $192 \times 192$. Though this does require some small amount of domain knowledge, it is computationally infeasible to process ImageNet (Deng et al., 2009) images at their raw resolution which varies widely and would require a context length in the millions. To assess representation quality, we fine-tune on ImageNet-100 under two settings, one with augmentation where we apply mixup (Zhang et al., 2017), cutmix (Yun et al., 2019), solarization, gaussian blur, grayscale, and color jitter, and one without any augmentations applied during fine-tuning.

We present ImageNet-100 results in Table 5. We find that our self-guided masking method performs well both with and without augmentation compared to the three baselines of (1) No pre-training, (2) Random masking, and (3) Masking random $16 \times 16$ patches. We found that with well-tuned strong augmentations, the random masking model was capable of outperforming the patch masking model. This may be due to a combination of a few factors. First, the the random masking method is generally the best at learning positional embeddings, which is quite important in the image domain given the large set of positional embeddings. Second, we found the random masking model required a high drop path rate (Huang et al., 2016) 0.3 to perform well, which is not applied for either the patch masking method nor our learned masking method. We suspect the high drop path rate can somewhat mitigate the issue of a trivial solution involving interpolating nearby pixels as in conjunction with the high masking rate, it is unlikely for the adjacent pixels of a masked pixel to be recovered. Finally, it is quite possible that many of the representations that are learned through pre-training processes could be eventually recovered with well tuned augmentation since the pre-training and fine-tuning sets are the same for this setting (Touvron et al., 2022).

For this reason, we test fine-tuning without any augmentation and find the random masking model to perform worse than the other methods. In comparison, we find that our self-guided masking method is capable of learning both strong positional embeddings given that the generated mask shapes are irregular compared to image patches that are consistent static squares, and when the learned masks have consolidated into full contiguous regions, our model can learn complex image representations similar to the patch masking model.

## 6 CONCLUSION

In this paper, we have presented a method, Self-Guided Masked Autoencoders (SMA), for learning representations from unlabeled data without any prior knowledge about our input. By leveraging large-scale pre-training, our approach alleviates the need for hand-designed masks and inductive priors, making it especially effective in domains without well-studied priors. Our results across three scientific domains demonstrate that prior knowledge is not necessary to train competitive representation learning methods. In summary, our approach offers a promising direction for unsupervised representation learning without the need for domain-specific inductive priors.

ACKNOWLEDGEMENTS

We thank Eric Anthony Mitchell for advice on pre-training language models. We also thank Shirley Wu, other members of IRIS Lab, and anonymous reviewers for helpful discussions and feedback. The compute for this work was supported by a HAI Google cloud credit grant sponsored by Fei Fei Li. Chelsea Finn is a CIFAR fellow. This work was supported by NSF, KFAS, and ONR grant N00014-20-1-2675.

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

| Method | RMSE (lower is better ↓) |
|---|---|
| TabNet (Arik and Pfister, 2021) | 8.909 |
| SNN (Klambauer et al., 2017) | 8.895 |
| AutoInt (Song et al., 2019) | 8.882 |
| GrowNet (Badirli et al., 2020) | 8.827 |
| MLP (Gorishniy et al., 2021) | 8.853 |
| DCN2 (Wang et al., 2021) | 8.890 |
| NODE (Popov et al., 2019) | 8.784 |
| ResNet (Gorishniy et al., 2021) | 8.846 |
| CatBoost (Prokhorenkova et al., 2018) | 8.877 |
| XGBoost (Chen and Guestrin, 2016) | 8.947 |
| FT-T (Gorishniy et al., 2021) | 8.855 |
| Transformer Scratch | 8.809 |
| Transformer Random Masking | 8.762 |
| Transformer Guided Masking (Ours) | **8.695** |

Table 6: **Year prediction for audio features**.

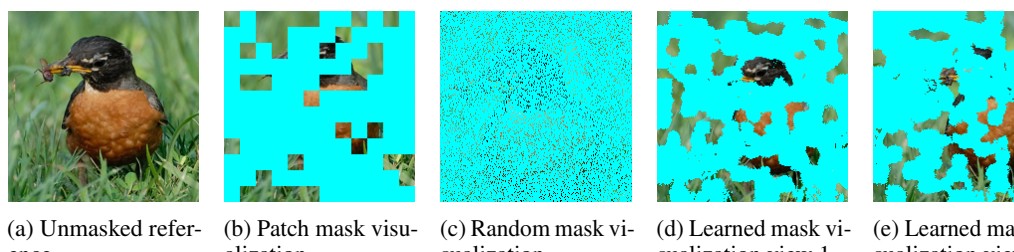

(a) Unmasked reference    (b) Patch mask visualization    (c) Random mask visualization    (d) Learned mask visualization view 1    (e) Learned mask visualization view 2

Figure 2: **Visualization of learned masks for ImageNet-100.** We show two sampled mask views generated from the same model weights to demonstrate our method can sample diverse masks even with static weights. We observe that masks are well clustered with respect to location and have a minor emphasis on color values.

| | |
|---|---|
| Unmasked Reference | Usually, he would be tearing around the living room, playing with his toys. But just one look at a minion sent him practically catatonic. |
| Random Masking | Usually, he would ▮e tea▮i▮g aroun▮ the living room, p▮▮yin▮ ▮ith hi▮ ▮oy▮. But jus▮ one look ▮ a minion sent h▮▮ ▮pra▮tical▮ ▮ catatonic. |
| Word Masking | Usually, ▮ would be ▮▮ around the ▮▮▮▮ room, playing with ▮ toys. But just ▮ look at a ▮▮▮ sent him practically catatonic. |
| Learned Mask View 1 | Usually, he would be tearing around the living room, playing with ▮ toys. B▮ just one look at a minion sent him ▮▮ti▮▮▮y catatonic. |
| Learned Mask View 2 | Usually, he would be tearing around the living room, playing with his toys▮ ▮▮ just one look at a minion sent him practically c▮▮onic. |

Table 7: **Visualization of learned masks on English Wikipedia + BooksCorpus.** We find that our method learns mask views that are clustered around specific positions with a minor focus on space delimiters. Actual sequences trained on are far generally far longer often nearly the max sequence size of 1024 therefore there may be some discrepancies in the actual masking ratio.

## A ADDITIONAL TABULAR RESULTS

In addition to the HIGGS tabular benchmark, we also benchmark on the year prediction from audio features tabular dataset (McFee et al., 2012). We use the same settings as Gorishniy et al. (2021) and report RMSE in Table 6.

## B ADDITIONAL IMPLEMENTATION DETAILS

For all experiments, we optimize using Adam Kingma and Ba (2014) with decoupled weight decay Loshchilov and Hutter (2017) and set $\beta_1 = 0.9$, $\beta_2 = 0.999$. Our other main hyperparameters are listed in Tables 8 to 12.

## C MASK VISUALIZATIONS

We visualize samples of masks generated for images in Figure 2 and text in Table 7. For each mask, we show two mask views generated using the same set of weights to demonstrate how our method can create diverse masks even with the same set of weights which is crucial to ensure the mask

| config | value |
|---|---|
| attention type | cross-attention |
| embed dimension | 512 |
| sequence length | 1024 |
| number of layers | 16 |
| qk dimension | 256 |
| qk heads | 8 |
| v dimension | 1024 |
| latents | 256 |

(a) Protein biology architecture

| config | random | guided |
|---|---|---|
| weight decay | 0.01 | |
| learning rate | 1e-4 | 3e-4 |
| epochs | 10 | |
| batch size | 256 | |
| masking rate | 0.2 | 0.15 |
| loss function | crossentropy | |

(b) Protein biology pre-training

| config | value |
|---|---|
| weight decay | 0.01 |
| learning rate | {1e-4, 5e-5} |
| epochs | 30 |
| batch size | {64, 128, 512} |

(c) Protein biology fine-tuning

Table 8: Protein Biology hyperparameters

prediction model does not memorize training samples. While our masks appear to be more biased toward positional as opposed to value information we suspect this is because the set of attention queries from which we generate attentions is static across all inputs. This means that the queries and thus attentions are calibrated to efficiently encode all inputs and therefore in the in this case positional information such as locality is generally more representative of input correlations across a whole dataset. For a similar reason previous Perceiver iterations Jaegle et al. (2021b) used repeat modules that cross attended the input again after processing the initial set of latent vectors, however applying our method to a subsequent set of cross attentions is expensive as it would require recomputing the entire block of layers again.

We find the generated image masks to be well clustered with respect to position which is expected given the high degree of correlation between a pixel token and it's adjacent values. Furthermore, due to our masking method we find that the generated masks form continuously masked regions which creates a more challenging prediction task requiring complex reasoning over unmasked tokens.

# D    RUNNING TIME INFORMATION

In developing our method, we found the repeated top-k operation challenging to parallelize because we must ensure each query masks unique tokens meaning the top-k operations need to be done sequentially to exclude the indices that were masked by other queries. This led to the approximation that we describe in Equation 5 in our main paper. In addition to alleviating the bottleneck of needing to compute the top-k operations iteratively, the approximation also improves the time complexity of the total top-k operations which accounts for a majority of the time in the masking process.

Suppose we select $m$ queries from which we compute $k$ inputs to mask for each from a total of $n$ inputs. Without our approximation, the average-case runtime complexity is $O(m(n + k))$ as we must perform m top-k operations. In comparison, with our method, the average case complexity is only $O(n + mk)$ because we just need to perform a single top-k operation with $mk$ selected values. Since m, the number of queries selected is generally a fraction of n, and it is always the case that $mk < n$ (we cannot mask more than or equal to 100% of inputs), our approximation reduces the complexity of top-k costs from squared to linear with respect to n (the number of inputs).

Empirically, the running time without our proposed approximation on the image domain is 1.13 steps/second. In comparison, the approximation training runs at 3.58 steps/second. Note that with the approximation, the time to compute masks does not significantly impact train step time (less than 2% of the time per step), but without the approximation, the time to compute masks becomes the bottleneck for large input counts. Additionally, empirically we found no difference in performance between the two methods across domains, and our approximation actually improves stability in our testing.

---

[6]For Higgs small experiments where the pre-train set is only 63k samples, we pretrain for 800 epochs.

| config | value |
|---|---|
| attention type | self-attention |
| sequence length | 256 |
| number of layers | 12 |
| qk dimension | 768 |
| qk heads | 12 |
| v dimension | 768 |

(a) Chemistry architecture

| config | value |
|---|---|
| weight decay | 0.01 |
| learning rate | 1e-4 |
| epochs | 30 |
| batch size | 128 |
| masking rate | 0.3 |
| loss function | crossentropy |

(b) Chemistry pre-training

| config | value |
|---|---|
| weight decay | 0.01 |
| learning rate | {1e-4, 3e-5} |
| epochs | 30 |
| batch size | 32 |

(c) Chemistry fine-tuning

Table 9: Chemistry hyperparameters

| config | value |
|---|---|
| attention type | self-attention |
| sequence length | 28 |
| number of layers | 4 |
| qk dimension | 128 |
| qk heads | 8 |
| v dimension | 128 |

(a) Particle physics architecture

| config | random | guided |
|---|---|---|
| weight decay | 0.01 | |
| learning rate | 1e-4 | |
| epochs | 50 | |
| batch size | 256 | |
| masking rate | 0.5 | 0.2 |
| loss function | mean-squared-error | |

(b) Particle physics pre-training

| config | value |
|---|---|
| weight decay | 1e-5 |
| learning rate | 3e-5 |
| epochs | 100 |
| batch size | 128 |

(c) Particle physics fine-tuning

Table 10: Particle physics hyperparameters

| config | value |
|---|---|
| attention type | cross-attention |
| embed dimension | 196 |
| sequence length | 36864 |
| number of layers | 8 |
| qk dimension | 768 |
| qk heads | 8 |
| v dimension | 768 |
| latents | 512 |

(a) Image architecture

| config | random | guided | patch |
|---|---|---|---|
| weight decay | 0.01 | | |
| learning rate | 1e-4 | | |
| epochs | 200 | | |
| batch size | 128 | | |
| masking rate | 0.85 | 0.75 | 0.75 |
| loss function | mean-squared error | | |

(b) Image pre-training

| config | pretrained | scratch |
|---|---|---|
| weight decay | 0.01 | |
| learning rate | 1e-4 | |
| epochs | 200 | 300 |
| batch size | 128 | |

(c) Image fine-tuning

Table 11: Image hyperparameters

| config | value |
|---|---|
| attention type | cross-attention |
| embed dimension | 512 |
| sequence length | 1024 |
| number of layers | 16 |
| qk dimension | 256 |
| qk heads | 8 |
| v dimension | 1024 |
| latents | 256 |

(a) Natural language architecture

| config | random | guided | word |
|---|---|---|---|
| weight decay | 0.01 | | |
| learning rate | 1e-4 | | |
| epochs | 10 | | |
| batch size | 256 | | |
| masking rate | 0.2 | 0.15 | 0.15 |
| loss function | crossentropy | | |

(b) Natural language pre-training

| config | value |
|---|---|
| weight decay | 0.01 |
| learning rate | 2e-5 |
| epochs | 30 |
| batch size | 32 |

(c) Natural language fine-tuning

Table 12: Natural language hyperparameters

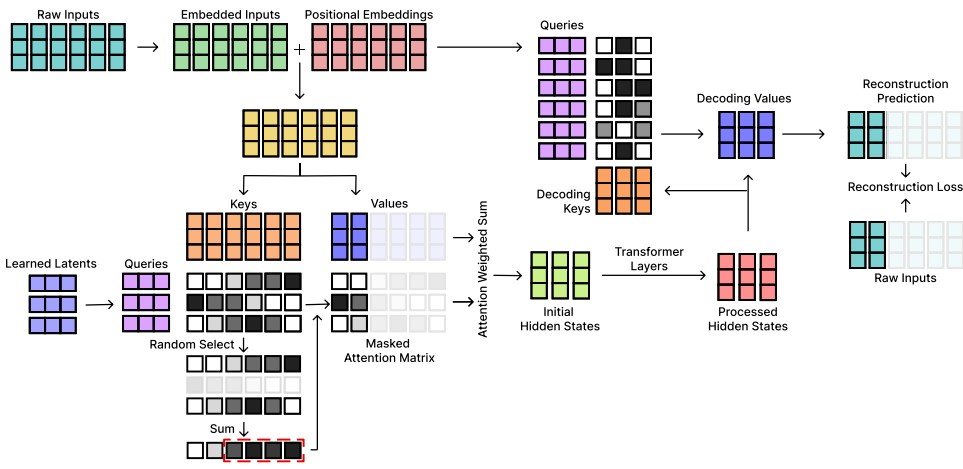

Figure 3: End-to-end diagram of SMA

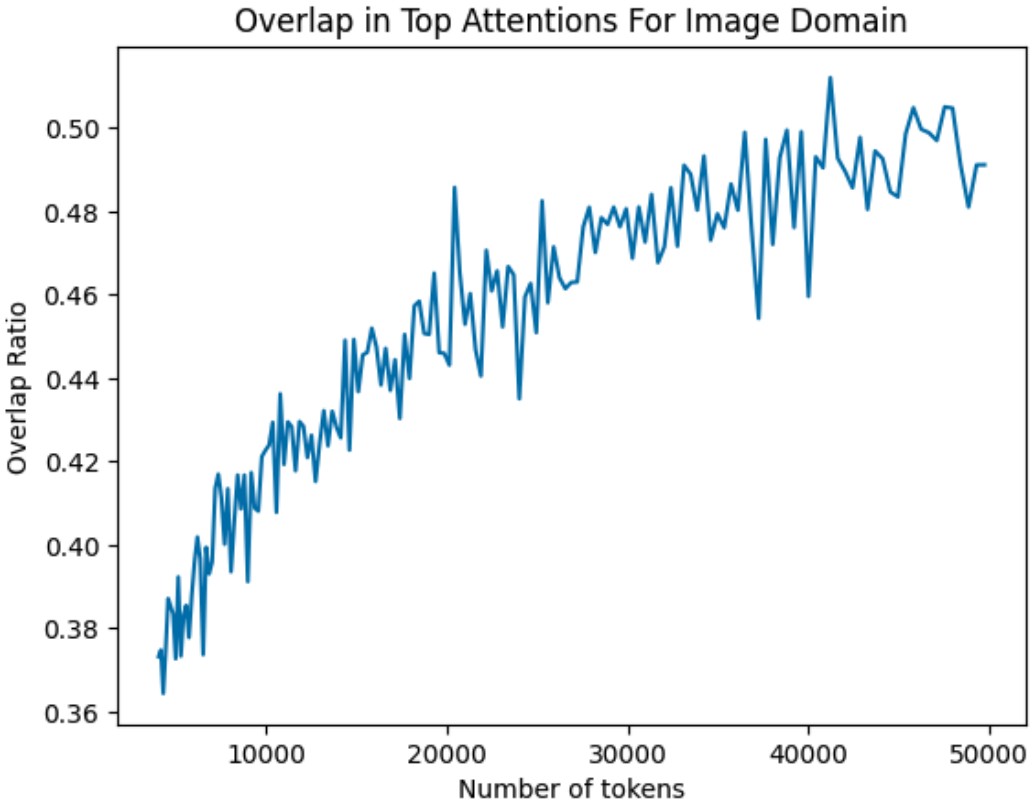

Figure 4: Relationship between number of inputs and the top attention overlap ratio for images. We find as we increase the number of inputs, the overlap ratio increases as well.

