# OpenReview forum: "Self-Guided Masked Autoencoders for Domain-Agnostic Self-Supervised Learning"
_ICLR.cc/2024/Conference — ICLR 2024 poster_

### Official Review · Reviewer_eSva · 2023-10-27

**Soundness:** 3 good
**Presentation:** 3 good
**Contribution:** 3 good
**Rating:** 6
**Confidence:** 3

**Summary:**

The paper proposes a domain-agnostic masked modeling approach for MAE's in the context of self-supervised learning, which can operate across both self and cross-attention architectures as well as different domains. An interesting idea, with a coherent story, and partially promising results.

**Strengths:**

- The paper reads well, and has a coherent story.
- The idea is somewhat novel, to my understanding, even though the impact might be marginal compared to a random equivalent.
- Experiments are elaborate and diverse in different domains, with further supporting results in the appendix.

**Weaknesses:**

1) I am missing an end-2-end sketch of the proposed architecture. I understand the goal is to keep it somewhat agnostic to task, architecture (self/cross-attention etc), but essential components can still be sketched. Algorithm 1 (which appears by the way too early in the text) to some extent covers this, but not totally.
2) Am I right to assume (3) - (5) are a mathematical notation of Fig. 1? If so, this is one of those examples where math becomes more of a problem than a solution!
3) The improvement SMA offers in settings without strong priors (such as protein property prediction, and chemical property prediction) is marginal, which is fine. However, here a simple random masking pattern seems to work pretty well, and that can be an efficient natural choice which is difficult to argue against. (See Tables 1, 2, 3 and 5).
4) I think it would be useful to have some reflections on the computational complexity (time and/or model space) of SMA. Is it the reason why ImageNet-100 is chosen instead of ImageNet itself?

5) The paper can benefit from another proof read; e.g. here are some minor suggestions:

    a) On page 2: These architectures demonstrates … => Demonstrate

    b) On page 3: Formally for an unlabeled … => Formally,

    On Page 4: the inputs of a single "group," => “group”,

    On Page 9: also use resize image … ? Doesn’t read well.

    And several others.

6) Other suggestions:

    a) I wouldn’t use capital letters for sets (but \mathcal{}) to avoid confusion with a constant value.

    b) I would avoid using two different l’s for the loss function and for query dimensions.

    c) Clarify the notation of mixed floor/ceiling in (3), in the subsequent paragraph.

   d) Please refer to table numbers in the text, this way the reader has to look for table content to associate it with descriptions. Table 5 is all of a sudden cited in the text!

    e) Even though self-explanatory to some extent, the algorithm is left unreferenced and unexplained.

**Questions:**

1) One can argue masking highlight correlated areas on the input actually poses an easier job to the prediction model no?

2) Why would repeated top-k(.) operations be too complex or hard to parallelize for different tokens? And why would (3) help to elevate this bottleneck?

3) In Table 1,  why shouldn’t one pick random masking? It seems to function as good as SMA, and the prior art, no? It’s way more efficient as well.

---

> ### Author Response · Authors · 2023-11-16
> **Response to Reviewer eSva (1/2)**
>
> Thank you for your review. We include responses to individual points below. If there are any remaining questions, please let us know.
>
> > I am missing an end-2-end sketch of the proposed architecture. I understand the goal is to keep it somewhat agnostic to task, architecture (self/cross-attention etc), but essential components can still be sketched. Algorithm 1 (which appears by the way too early in the text) to some extent covers this, but not totally.
>
> For completeness, we have included an additional full visualization of the architecture in our supplementary materials (Fig 3). We hope this clarifies any remaining questions regarding the end-to-end structure.
>
> > Am I right to assume (3) - (5) are a mathematical notation of Fig. 1? If so, this is one of those examples where math becomes more of a problem than a solution!
>
> While we agree that the figure is easier to understand as it conveys the core mechanism in a more intuitive way, we believe that including a formal definition is beneficial for completeness and clarity.
>
> > The improvement SMA offers in settings without strong priors (such as protein property prediction and chemical property prediction) is marginal, which is fine. However, here a simple random masking pattern seems to work pretty well, and that can be an efficient natural choice which is difficult to argue against. (See Tables 1, 2, 3 and 5).
>
> Our method improves over the random masking method in every benchmark we have tested without any drawbacks, as the per-step training time is the same. Therefore we believe our method is comprehensively better than the random masking method. Additionally, the amount of improvement over random masking is closely tied to how much redundancy exists in the domain itself, with protein biology likely having the least redundancy in discrete protein tokens.
>
> > I think it would be useful to have some reflections on the computational complexity (time and/or model space) of SMA. Is it the reason why ImageNet-100 is chosen instead of ImageNet itself?
>
> The main purpose of the ImageNet100 experiments is to provide a comparison to hand-designed masks in a controlled setting and not to suggest our method is state-of-the-art for image pretraining. We do acknowledge that the Perceiver architecture (our method does not actually introduce any significant additional compute) on raw images still has a high computational cost. Compared to the original perceiver, our architecture requires approximately 1/10th of the FLOPs, but scaling to the full ImageNet dataset is beyond our computational resources. In fact, the Imagenet100 experiments still incurred the most computational cost out of all domains.
>
> > The paper can benefit from another proof read; e.g. here are some minor suggestions
>
> Thank you for the thorough editing suggestions. We have edited the paper to take all of these into account.
>
> > One can argue masking highlight correlated areas on the input actually poses an easier job to the prediction model no?
>
> Empirically we find our method to have a higher reconstruction loss compared to random masking despite lower masking rates across modalities indicating that our method generates a more challenging masking objective.

---

> > ### Comment · Reviewer_eSva · 2023-11-16
> > **Computational  and time complexity**
> >
> > Thanks for the response to my other remarks.
> >
> > You mention several interesting points regarding complexity (model & time), I cannot see any additions/adjustments in the revised version in that regard. I think reflection on this topic can be a good addition to the paper.

---

> ### Author Response · Authors · 2023-11-16
> **Response to Reviewer eSva (2/2)**
>
> > Why would repeated top-k(.) operations be too complex or hard to parallelize for different tokens? And why would (3) help to elevate this bottleneck?
>
> The repeated top-k operation is challenging to parallelize because we must ensure each query masks unique tokens meaning the top-k operations need to be done sequentially to exclude the indices that were masked by other queries.
>
> Regarding the complexity, suppose we select m queries from which we compute k inputs to mask for each from a total of n inputs. Without our approximation, the average-case runtime complexity is O(m(n + k)) as we must perform m top-k operations. In comparison, with our method, the average case complexity is only O(n + mk) because we just need to perform a single top-k operation with mk selected values. Since m, the number of queries selected is generally a fraction of n, and it is always the case that mk < n (we cannot mask more than or equal to 100% of inputs), our approximation reduces the complexity of top-k costs from squared to linear with respect to n (the number of inputs).
>
> Empirically, the running time without our proposed approximation on the image domain is 1.13 steps/second. In comparison, the approximation training runs at 3.58 steps/second. Note that with the approximation, the time to compute masks does not significantly impact train step time (less than 2% of the time per step), but without the approximation, the time to compute masks becomes the bottleneck for large input counts. Additionally, empirically we found no difference in performance between the two methods across domains, and our approximation actually improves stability in our testing.
>
> > In Table 1, why shouldn’t one pick random masking? It seems to function as good as SMA, and the prior art, no? It’s way more efficient as well.
>
> Random masking is actually not more efficient by a noticeable amount, as even in the image modality where the difference in runtime is greatest, our algorithm is only 1% slower in training steps/second (3.63 steps/second vs 3.58 steps/second). Secondly, our method surpasses the random masking performance across all benchmarks. We suspect the difference is relatively smaller in the protein biology domain as the genomics data is somewhat preprocessed from its raw form by the benchmark.

---

> > ### Comment · Reviewer_eSva · 2023-11-16
> > **Convincing response, further actions.**
> >
> > Thanks for the top-k(.) explanations, makes more sense now.
> >
> > I'm still not convinced why one shouldn't pick a random strategy when (i) it's domain agnostic and requires no design optimization, (ii) the performance margin of SMA beyond random is only a couple of percent (or even less) in all settings, (iii) random is 1% faster? In which setting SMA offers substantial improvement beyond random?

---

> ### Author Response · Authors · 2023-11-17
> **Response to Comparison with random masking**
>
> > I'm still not convinced why one shouldn't pick a random strategy when (i) it's domain agnostic and requires no design optimization, (ii) the performance margin of SMA beyond random is only a couple of percent (or even less) in all settings, (iii) random is 1% faster? In which setting SMA offers substantial improvement beyond random?
>
> Firstly, we do believe that these improvements are quite substantial as evaluation by finetuning tends to decrease the difference between methods. For example in the original masked autoencoders paper [1] the difference between training from scratch and using MAE pre training was only 2-3%.
>
> Furthermore, the benefit of random masking largely comes from learning positional embeddings which need to be learned from scratch in our setting. Therefore, it is easiest to observe the shortcomings of random masking in a domain such as point cloud representation learning. To create point clouds, random points are sampled from 3D object representations. For this domain, the random masking objective is ill-defined because any randomly masked point does not change the structure of the pointcloud enough to suggest a missing part of the object as it is possible that point was simply not sampled. In comparison our method masks whole parts of the object and is capable of generating a rich, well-defined self-supervision task. We update our paper appendix with a visualization of a mask generated by our method to demonstrate this point.
>
> Empirically, we perform an experiment by pre training on ModelNet40 followed by finetuning on ModelNet40 classification. In this setting, SMA provides a 1.9% boost in classification accuracy compared to training from scratch while pre-training with random masking leads to the loss diverging due to the poorly defined self-supervisory task. We hope this example provides a clear case where random masking cannot work and SMA is necessary to provide any improvement over from scratch training. Note that these results were produced before the response period, but they were not included in our paper as we originally were targeting scientific domain applications.
>
> [1] K. He, X. Chen, S. Xie, Y. Li, P. Dollár, and R. Girshick. Masked autoencoders are scalable vision learners. In Proceedings of the IEEE/CVF Conference on Computer Vision and Pattern Recognition, pages 16000–16009, 2022.

---

> ### Author Response · Authors · 2023-11-17
> **Added Running Time Information to Supplementary Materials**
>
> > Thanks for the response to my other remarks. You mention several interesting points regarding complexity (model & time), I cannot see any additions/adjustments in the revised version in that regard. I think reflection on this topic can be a good addition to the paper.
>
> Thank you for the suggestion. We have added the analysis and explanation to the appendix as our main paper is very near the length limit as is.

---

> ### Author Response · Authors · 2023-11-20
> **Checking In**
>
> We wanted to follow up to see if the response and revisions address your concerns. We would be happy to provide further clarifications and revisions if you have any more questions and if not we would greatly appreciate it if you would reevaluate our paper score. Thank you again for your reviews which helped to improve our paper!

---

> > ### Comment · Reviewer_eSva · 2023-11-21
> > **Raising score**
> >
> > Thanks for the reminder, I am going to raise my score.

---

### Official Review · Reviewer_Md4o · 2023-10-28

**Soundness:** 3 good
**Presentation:** 3 good
**Contribution:** 3 good
**Rating:** 6
**Confidence:** 3

**Summary:**

The paper introduces the Self-guided Masked Autoencoders (SMA), a domain-agnostic self-supervised learning technique based on masked modeling. Distinguishing itself from traditional self-supervised methods, which often incorporate domain-specific knowledge, the SMA refrains from using any form of tokenizer or making assumptions about the structure of raw inputs. Instead, it dynamically computes masks based on the attention map of the initial encoding layer during masked prediction training. The authors demonstrate SMA's effectiveness across three diverse domains: protein biology, chemical property prediction, and particle physics, where it achieves state-of-the-art performance without relying on domain-specific expertise.

**Strengths:**

S1 - Domain-Agnostic:

SMA is designed to be entirely domain-agnostic, ensuring it can be applied widely without needing domain-specific adjustments.

S2 - Dynamic Mask Learning:

Rather than depending on fixed tokenizers or pre-defined masking strategies, SMA innovatively learns the relationships between raw inputs to determine useful mask sampling.

S3 - Decent Performance on several datasets: On all evaluated benchmarks, SMA not only competes but surpasses the state-of-the-art, indicating its potential as a leading approach in self-supervised learning.

**Weaknesses:**

W1 Experiments:

While the authors report the results of ImageNet100, the results on the full dataset are also expected to ensure a comprehensive evaluation. Additionally, I'm also curious about the pre-trained encoder's performance on segmentation and object detection tasks.

For tabular datasets like HIGGS, the results are promising. However, I'd suggest authors extend the work to broader tabular datasets as the performance of deep learning-based models may vary a lot. Additional experiments are not super expensive in this domain but will give a more comprehensive evaluation.

 W2 Training and Inference Efficiency:

Authors in this paper claim that the proposed feature space masking is efficient. However, unless I missed it, I failed to see related statistical results/analysis to prove such a claim.

**Questions:**

The main questions are listed in Weaknesses. I'd raise my score if they were appropriately addressed.

---

> ### Author Response · Authors · 2023-11-16
> **Response to Reviewer Md4o**
>
> Thank you for your review. We include responses to each of your questions below. If there are any remaining questions, please let us know.
>
> > While the authors report the results of ImageNet100, the results on the full dataset are also expected to ensure a comprehensive evaluation.
>
> Unfortunately, scaling to the full ImageNet dataset is beyond our computational resources. In fact, the Imagenet100 experiments still incurred the most computational cost out of all domains due to the high number of input tokens per image (36,864). We even downscaled the original Perceiver architecture to approximately 1/10th the size in terms of FLOPs.
>
> >Additionally, I'm also curious about the pre-trained encoder's performance on segmentation and object detection tasks.
>
> Object detection is challenging given the time constraints since we are not aware of any prior works using Perceiver for object detection and testing and tuning a new architecture is unfortunately not possible during the response period. In the case of semantic segmentation, we are aware of a prior Perceiver work [1] which demonstrate semantic segmentation results on PASCAL VOC. However, their architecture used a resolution of 512 x 512 whereas ours uses one of size 192 x 192. Additionally theirs is much larger in terms of number of latents and hidden state dimension and is pretrained over ImageNet-1k. Therefore it would be challenging to pretrain this new larger architecture during the response period. Lastly, the purpose of the image domain experiments is not to suggest our pretrained model is state-of-the-art, but rather to just compare our guided masking method against a domain-specific masking method (patch masking) which we outperform.
>
> > Authors in this paper claim that the proposed feature space masking is efficient. However, unless I missed it, I failed to see related statistical results/analysis to prove such a claim.
>
> In terms of additional operations, our proposed masking method requires only a single additional summation, softmax, and top-k operation. The reason our method is efficient is that we mask using the first-layer attention mask meaning there is no additional computation of any layer outputs.
>
> Empirically, the per step time of our method in the image domain is 3.58 steps/second for our guided masking method (3.63 steps/second for random masking and 3.54 steps/second for patch masking). The overhead cost of generating masks accounts for less than 2% of the time per training step, even in the image domain, where the large number of inputs magnifies any costs.
>
> We have added further details on this in the supplementary materials of our paper as well.
>
> > For tabular datasets like HIGGS, the results are promising. However, I'd suggest authors extend the work to broader tabular datasets as the performance of deep learning-based models may vary a lot. Additional experiments are not super expensive in this domain but will give a more comprehensive evaluation.
>
> Previously we used the HIGGS tabular dataset as we were focused on scientific applications (in line with the chemistry and biology domains we used). However, we agree it would make the evaluation more comprehensive with more tabular results. We have benchmarked 1 more tabular dataset, the year prediction regression benchmark, to now have both a classification and regression task. We have rendered the table below and added it to the supplementary materials of our paper as well.
>
> | Method                                 | RMSE (lower is better ↓) |
> |----------------------------------------|---------------------------|
> | TabNet                                | 8.909                     |
> | SNN                                   | 8.895                     |
> | AutoInt                               | 8.882                     |
> | GrowNet                               | 8.827                     |
> | MLP                                   | 8.853                     |
> | DCN2                                  | 8.890                     |
> | NODE                                  | 8.784                     |
> | ResNet                                | 8.846                     |
> | CatBoost                              | 8.877                     |
> | XGBoost                               | 8.947                     |
> | FT-T                                  | 8.855                     |
> | Transformer Scratch                   | 8.809                     |
> | Transformer Random Masking             | 8.762                     |
> | **Transformer Guided Masking (Ours)**  | **8.695**                 |
>
> [1] J. Carreira, S. Koppula, D. Zoran, A. Recasens, C. Ionescu, O. Henaff, E. Shelhamer, R. Arandjelovic,
> M. Botvinick, O. Vinyals, et al. Hierarchical perceiver. arXiv preprint arXiv:2202.10890, 2022

---

> ### Author Response · Authors · 2023-11-20
> **Checking In**
>
> We wanted to follow up to see if the response and revisions address your concerns. We would be happy to provide further clarifications and revisions if you have any more questions and if not we would greatly appreciate it if you would reevaluate our paper score. Thank you again for your reviews which helped to improve our paper!

---

> ### Author Response · Authors · 2023-11-22
> **Following up**
>
> Thanks again for your review. We wanted to follow up again to make sure that your concerns are being properly addressed. Please let us know if you have additional questions. If all your concerns have been resolved, we would greatly appreciate it if you could reconsider and adjust your rating and evaluation of our work.

---

> > ### Comment · Reviewer_Md4o · 2023-11-23
> >
> > I appreciate the authors' reply. I've also read other reviewers' comments. I'd raise the score from my side, given the additional experience.

---

### Official Review · Reviewer_ijD8 · 2023-10-28

**Soundness:** 3 good
**Presentation:** 3 good
**Contribution:** 3 good
**Rating:** 6
**Confidence:** 4

**Summary:**

This work tackles the problem of domain-agnostic self-supervised learning which does not assume any prior knowledge about the domain itself. The authors propose Self-Guided Masked Autoencoders (SMA) that computes masks based on the attention map of the the model at the first layer. SMA is shown to outperform random masking and other domain-specific baselines on a wide-range of tasks including protein/chemical property prediction, particle physics classification and natural language tasks.

**Strengths:**

- The problem of domain-agnostic self-supervised learning is an important problem given it’s wide applicability particularly in deep learning for science. The proposed method is simple and elegant.
- The authors show results on a wide-variety of domains with impressive performance without any domain knowledge. It is interesting to see that the results with SMA (without any domain-specific tokenizers) are comparable/better than other methods with domain knowledge.

**Weaknesses:**

- Missing important related work:
    - DABS [1, 2] is a benchmark for domain agnostic self-supervised learning algorithms. This benchmark consists of semiconductor wafers, multispectral satellite imagery, protein biology, bacterial genomics, particle physics, Speech Recordings, Chest Xrays. DABS also has baselines in the form of Generalized masked autoencoding, Capri (Hybrid Masked-Contrastive Algorithm), e-Mix and Shuffled Embedding Detection (ShED). Demonstrating the effectiveness of SMA on this benchmark would strengthen the paper. I understand that the authors have already shown results on particle physics and protein datasets but comparing with these baselines would lead to a more complete results section. The authors can also discuss and compare SMA with these baselines.
    - The authors should compare and contrast with related literature [3, 4, 5].
- The authors can run some ablations to better understand the proposed method, SMA. For instance, how does the masking ratio impact performance in various domains? It may be interesting to analyze the performance if the masks are computed in second/ third layer (or any kth layer) instead of first layer in all the experiments.


[1] Tamkin, Alex, et al. "DABS: a Domain-Agnostic Benchmark for Self-Supervised Learning." *Thirty-fifth Conference on Neural Information Processing Systems Datasets and Benchmarks Track (Round 1)*. 2021.

[2] Tamkin, Alex, et al. "DABS 2.0: Improved datasets and algorithms for universal self-supervision." *Advances in Neural Information Processing Systems* 35 (2022): 38358-38372.

[3] Wu, Huimin, et al. "Randomized Quantization: A Generic Augmentation for Data Agnostic Self-supervised Learning." *Proceedings of the IEEE/CVF International Conference on Computer Vision*. 2023.

[4] Lee, Kibok, et al. "i-mix: A domain-agnostic strategy for contrastive representation learning." arXiv preprint arXiv:2010.08887 (2020).

[5] Verma, Vikas, et al. "Towards domain-agnostic contrastive learning." International Conference on Machine Learning. PMLR, 2021.

**Questions:**

1. What are the hyperparameters in SMA? Is masking ratio the only hyperparameter? Can the authors explain why certain domains require a high masking rate compared to others (as mentioned in Table 7 to 11)
2. How would the performance differ if a domain-specific tokenizer is used in Chemical property prediction?

---

> ### Author Response · Authors · 2023-11-16
> **Response to Reviewer ijD8 (1/2)**
>
> Thank you for your thoughtful feedback. We address your considerations below. Please let us know if you have any remaining questions or concerns.
>
> > DABS [1, 2] is a benchmark for domain agnostic self-supervised learning algorithms. This benchmark consists of semiconductor wafers, multispectral satellite imagery, protein biology, bacterial genomics, particle physics, Speech Recordings, Chest Xrays. DABS also has baselines in the form of Generalized masked autoencoding, Capri (Hybrid Masked-Contrastive Algorithm), e-Mix and Shuffled Embedding Detection (ShED). Demonstrating the effectiveness of SMA on this benchmark would strengthen the paper. I understand that the authors have already shown results on particle physics and protein datasets but comparing with these baselines would lead to a more complete results section. The authors can also discuss and compare SMA with these baselines.
>
> The main reason we do not use the DABS benchmark is that their definition of domain agnostic slightly differs from ours, as their benchmark allows for domain-specific tokenization, which is one of the main processes we aim to avoid with our method. As a result, we believe using their benchmark would create some confusion around our definition of domain-agnostic especially because there is some overlap with tasks and datasets used (we have updated the paper to acknowledge these similarities with citations).
>
> However, we agree it is important to have other domain-agnostic baselines to compare against, and we have added results comparing against [5], the current state-of-the-art in domain-agnostic representation learning (far outperforming the e-Mix method as an augmentation method and to our knowledge state-of-the-art on DABS). We significantly outperform their method and include the table of results below, along with updating the original ImageNet100 results in our paper.
>
> | Architecture                             | Pretraining Method                                  | Domain Agnostic | w/o augmentation | w/ augmentation |
> |-------------------------------------------|------------------------------------------------------|------------------|-------------------|------------------|
> | **Perceiver**                            | Patch                                                | ❌              | **79.6**          | 84.3             |
> | **ResNet**                                | None                                                 | ✔️             | 72.5              | 78.1             |
> | **ResNet**                                | Random Quantization [6] + MoCo-v3                         | ✔️                 | 76.9              | 81.6             |
> | **Perceiver**                            | None                                                 | ✔️             | 52.5              | 62.7             |
> | **Perceiver**                            | Random                                               | ✔️                 | 77.6              | 85.6             |
> | **Perceiver**                            | SMA (ours)                                           | ✔️                 | **78.5**          | **86.1**         |
>
> Despite their method using a ResNet-50 architecture which outperforms our Perceiver Architecture when trained from scratch on ImageNet-100, our method still outperforms this prior art, we add the results from their method to our other ImageNet100 results. Also note that when running their method on our same Perceiver architecture, we found it resulted in embedding collapse, which we theorize is due to the scratch initialization of positional embeddings, and the contrastive loss does not facilitate the learning of these positional embeddings well.  We include a rendering of the updated Imagenet100 table below for quick reference. We apply the method of [5] with the MoCo-v3 framework, as the original paper found MoCo-v3 to work best with their proposed augmentation. Technically the ResNet architecture is not domain-agnostic, but we denote their method as domain-agnostic because their proposed augmentation can be done domain-agnostically.
>
> Additionally, we now compare against [6], which explores tabular pretraining methods in the same setting as our HIGGS Small benchmark. The main difference is they use the ROC-AUC metric for the HIGGS Small subset achieving a best score of 82.1 ROC-AUC. In comparison, our scratch model achieves an ROC-AUC score of 80.78, our random masking model achieves a score of 82.0 ROC-AUC, and our guided masking model achieves a score of 82.8. Therefore, our method remains the best in this setting.

---

> ### Author Response · Authors · 2023-11-16
> **Response to Reviewer ijD8 (2/2)**
>
> > The authors should compare and contrast with related literature [3, 4, 5].
>
> We have compared against [5], which has the best previous empirical results. See our response above for comparisons on the image domain.
>
> > The authors can run some ablations to better understand the proposed method, SMA. For instance, how does the masking ratio impact performance in various domains? It may be interesting to analyze the performance if the masks are computed in second/ third layer (or any kth layer) instead of first layer in all the experiments.
>
> We are not entirely certain what is meant by computing masks in the second/third layer. In our architecture, the attention with the inputs is computed only once. While it could be computed again in theory by computing attention with inputs, computing masks later would require recomputing the initial hidden states again with masked inputs. In comparison with our formulation, no recomputation is necessary, which we believe is important as it allows our method to be applied without any overhead computation.
>
> > What are the hyperparameters in SMA? Is masking ratio the only hyperparameter? Can the authors explain why certain domains require a high masking rate compared to others (as mentioned in Table 7 to 11)
>
> Yes, masking ratio is the only additional hyperparameter that is tuned for each domain. Regarding the difference in masking rate across domains, this is still somewhat of an open question in masked modeling, but most works point to increased redundancy between different domains requiring different optimal masking ratios. We provide details on how masking ratio sweeps were done below.
>
> For NLP and vision domains we simply set the masking ratio of our method to the masking ratio of the hand-designed masks. For vision random masking we used the 0.85 value from the Perceiver paper, which performed an experiment with random pixel masking as well and already performed a sweep. For NLP random masking we performed a sweep of values from 0.15 to 0.3 at increments of 0.05. We observed a decrease in performance beyond 0.2 and therefore did not sweep higher.
>
> For the protein and chemistry domains, we swept from 0.15 to 0.3 and used the optimal value. Prior work has also found optimal masking values to be in the same range [1, 2]. For the HIGGS benchmark, we found further improvements for random masking at higher rates therefore, we swept until 0.5, where we eventually noticed worse performance beyond this masking rate.
>
> > How would the performance differ if a domain-specific tokenizer is used in Chemical property prediction?
>
> Our work aims to avoid using any domain-specific tokenizers. In the case of the chemistry domain, the domain-specific tokenizers simply reduce certain element symbols in SMILES strings from many characters into a single token. While the results may vary depending on the domain-specific tokenizer used, from our testing with the ChemBERTa-2 tokenizer we found the difference in performance to be statistically insignificant on downstream tasks. Additionally we found the tokenizer to only merge approximately 2% of the tokens on the pre-training set meaning the difference between their tokenizer and a simple UTF-8 mapping (the one we use) is relatively small.
>
> [5] Wu, Huimin, et al. "Randomized Quantization: A Generic Augmentation for Data Agnostic Self-supervised Learning." Proceedings of the IEEE/CVF International Conference on Computer Vision. 2023.
>
> [6] I. Rubachev, A. Alekberov, Y. Gorishniy, and A. Babenko. Revisiting pretraining objectives for tabular deep learning. arXiv preprint arXiv:2207.03208, 2022.

---

> ### Author Response · Authors · 2023-11-20
> **Checking In**
>
> We wanted to follow up to see if the response and revisions address your concerns. We would be happy to provide further clarifications and revisions if you have any more questions and if not we would greatly appreciate it if you would reevaluate our paper score. Thank you again for your reviews which helped to improve our paper!

---

> ### Author Response · Authors · 2023-11-22
> **Following up**
>
> Thanks again for your review. We wanted to follow up again to make sure that your concerns are being properly addressed. Please let us know if you have additional questions. If all your concerns have been resolved, we would greatly appreciate it if you could reconsider and adjust your rating and evaluation of our work.

---

> ### Comment · Reviewer_ijD8 · 2023-11-22
> **Response to Reviewer**
>
> I thank the authors for the detailed rebuttal. I appreciate the pointwise responses and additional experiments. I also apologize for the delayed response.
>
> Comparison with Random Quantization method on ImageNet-100 is great. I still feel that the authors can compare with other baselines in DABS 2.0 i.e e-Mix on all domains (The random quantization results are on ImageNet-100 only).
>
> Given that most of my concerns are addressed, I will increase my score accordingly.

---

### Official Review · Reviewer_zapB · 2023-10-31

**Soundness:** 2 fair
**Presentation:** 3 good
**Contribution:** 3 good
**Rating:** 6
**Confidence:** 3

**Summary:**

This paper proposes a new mechanism called Self-Guided Masked Autoencoders (SMA), that acts as a generic masking procedure in the embedding space, and is therefore agnostic to the nature of the input data. SMA is evaluated on a wide variety of tasks ranging from molecular and chemical property prediction tasks, to image classification and NLP tasks. A reasonable level of performance is demonstrated without domain-specific data augmentations throughout the tasks.

**Strengths:**

1) The idea of generic masking in the self-attention layers directly is novel and promising. Generic masking was explored previously with the data2vec models but here the mechanism seems to be more principled and applicable to any domain using transformers.

2) The initial results on various tasks without domain-specific data augmentation is encouraging and might lead with further exploration to general and principled architectures for self-supervised learning on any kind of data.

3) The results on biology and chemistry tasks are convincing and competitive with prior work.

**Weaknesses:**

1) The results on image classification and NLP tasks are only on toy datasets and seem to be too preliminary to convince people from these communities to try the approach. The gains are marginal and only a small set of methods are compared. The final model is far from the state-of-the-art in NLP and vision. I would recommend if possible to be more ambitious and demonstrate results on more large scale tasks such as linear evaluation on ImageNet.

2) Some design choices are not well motivated and should be ablated properly. For example masking ratios, number k of elements in Eq.3

**Questions:**

1) How do you tune the masking parameters, such as the number of queries and input to mask ? (masking ratio) How difficult is it ?

2) Do you need to mask queries ? Could you simply mask the input randomly ? Could you clarify if this corresponds to “Random Masking” in your tables ?

---

> ### Author Response · Authors · 2023-11-16
> **Response to Reviewer zapB**
>
> Thank you for your thoughtful feedback. We address your comments below. Please let us know if you have any remaining questions or concerns.
>
> > The results on image classification and NLP tasks are only on toy datasets and seem to be too preliminary to convince people from these communities to try the approach. The gains are marginal and only a small set of methods are compared. The final model is far from the state-of-the-art in NLP and vision.
>
> We agree with this assessment. The purpose of these experiments was not to suggest our method be used for the NLP or vision modalities, as these two domains have well-developed inductive biases such as causality in NLP or augmentations in vision.  Rather, these experiments showcase how our method compares with hand-designed masks in a controlled setting, given that the other domains we benchmark on do not have hand-designed masks. Our method is most useful in domains without well-developed inductive biases and allows for competitive self-supervised representation learning without domain knowledge.
>
> > I would recommend if possible to be more ambitious and demonstrate results on more large scale tasks such as linear evaluation on ImageNet
>
> We note that our purpose is _not_ to advance the state of the art in established domains such as language or images but to develop a domain-agnostic method that readily applies to new domains such as protein biology or particle physics. Currently, scaling to the full ImageNet dataset is beyond our computational resources. In fact, even at ImageNet100, the image experiments are the most costly in terms of compute, and an experiment with ImageNet-1k would require at least 10x the computational resources.
>
> > Some design choices are not well motivated and should be ablated properly. For example masking ratios, number k of elements in Eq.3
>
> Following prior work we sweep for the optimal masking ratio in each of our domains. The exceptions are in the language and image domains we do not sweep for patch masking or word masking, given these masking ratios have been well studied. The number of elements $k$ is dependent on the masking ratio and is always 2 * # of inputs / # of queries. Generally, we found this to work well across domains, and we try to minimize the need to sweep for hyperparameters with the exception of masking ratio.
>
> > How do you tune the masking parameters, such as the number of queries and input to mask ? (masking ratio) How difficult is it ?
>
> For NLP and vision domains, we simply set the masking ratio of our method to the masking ratio of the hand-designed masks. For vision random masking, we used the 0.85 value from the Perceiver paper, which performed an experiment with random pixel masking as well and already performed a sweep. For NLP random masking, we performed a sweep of values from 0.15 to 0.3 at increments of 0.05. We observed a decrease in performance beyond 0.2 and therefore did not sweep higher.
> For the protein and chemistry domains, we swept from 0.15 to 0.3 and used the optimal value. Prior work has also found optimal masking values to be in the same range [1, 2]. For the HIGGS benchmark, we found further improvements for random masking at higher rates therefore, we swept until 0.5, where we eventually noticed worse performance beyond this masking rate.
>
> > Do you need to mask queries ? Could you simply mask the input randomly ? Could you clarify if this corresponds to “Random Masking” in your tables ?
>
> Yes, “simply masking the input randomly” is the random masking baseline in our tables. We find that we outperform this baseline in every domain with our method not introducing any additional compute. Lastly, we would like to clarify that we are not masking the queries but rather using the query attention to calculate an optimal mask of the raw input values.

---

> ### Author Response · Authors · 2023-11-20
> **Checking In**
>
> We wanted to follow up to see if the response and revisions address your concerns. We would be happy to provide further clarifications and revisions if you have any more questions and if not we would greatly appreciate it if you would reevaluate our paper score. Thank you again for your reviews which helped to improve our paper!

---

### Official Review · Reviewer_36zB · 2023-11-01

**Soundness:** 3 good
**Presentation:** 2 fair
**Contribution:** 3 good
**Rating:** 6
**Confidence:** 2

**Summary:**

This paper presented a Self-Guided Masked Autoencoder for domain-agnostic SSL, with experiments largely focusing on data from scientific domains such as biology, chemistry, and physics. It selects tokens to mask via attention maps from the first encoding layer (either self-attention or cross-attention) and masks the tokens with high aggregated attention weights. The authors argue that such an approach masks highly correlated semantic regions regardless of domain priors. The authors show strong results in protein biology, chemical property prediction, and particle physics.

This paper works on an interesting topic with great potential impact and it is clearly written; however, the paper slightly lacks quality, therefore the reviewer recommends a borderline rejection.

**Strengths:**

Originality: the method is inspired by Perceiver (Jaegle 2021b) and adapted the latent query technique to the attention module in masked autoencoder. Although the method is not particularly novel, the reviewer reckons that this is the first work to improve MAE’s domain-agnostic property via attention-map-based mask selection. There are other mask selection works, but they are primarily domain-specific [1, 2].

Clarity: the paper has a good flow and is, in general, easy to read.

Significance: domain-agnostic SSL is an important research topic as the community is seeing the merging of multi-domain, multi-modality pretraining. This paper serves as a nice step forward in this direction by using a generic attention-based mask selection technique for MAE pre-training.

[1] Li, Gang, et al. "Semmae: Semantic-guided masking for learning masked autoencoders." Advances in Neural Information Processing Systems 35 (2022): 14290-14302.

[2] Wilf, Alex, et al. "Difference-Masking: Choosing What to Mask in Continued Pretraining." arXiv preprint arXiv:2305.14577 (2023).

**Weaknesses:**

Post-rebuttal update: the responses include new experimental comparisons and successfully address all of the reviewer's concerns. Therefore, the reviewer updated the rating.

------

Originality: the paper did not cite or compare other domain-agnostic SSL methods, either contrastive [3] or masking [4]. Also, the key components, latent query tokens (Jaegle 2021b) and the KeepTopK (Shazeer et al. 2017) are not novel, further weakening the originality of this work.

Quality: the quality of this work is lacking. Empirical performance improvement can be limited, such as results for the MoleculeNet (Table 2), where the proposed method is sometimes worse than the baseline method (BACE and HIV of Uni-Mol-20M), or improvement can seem limited (Lipo). While a lower performance is common, since there are only limited baselines (TabNet for the HIGGS benchmark and baselines for the HIGGS benchmark are all from 2021 or prior) and this work is quite empirical, the performance difference can seem noticeable. Nevertheless, the reviewer is not familiar with the benchmarks and did not extensively search for new work with higher results, and will hugely benefit from a response from the authors explaining why the baselines are few.

Clarity: some parts of the paper seem confusing; the details are in the Questions section.

[3] Verma, Vikas, et al. "Towards domain-agnostic contrastive learning." International Conference on Machine Learning. PMLR, 2021.

[4] Yang, Haiyang, et al. "Domain invariant masked autoencoders for self-supervised learning from multi-domains." European Conference on Computer Vision. Cham: Springer Nature Switzerland, 2022.

**Questions:**

1. Page 5, “First, because the top attention tokens chosen for each query have significant overlap, we often do not actually mask the target masking ratio, and this issue worsens as $n$ increases.“
* How much overlap did the top attention tokens have? Did the authors quantify them? How does this issue worsen as $n$ increases – a log or linear relationship?

2. Page 5, “...while achieving the desired masking ratio and parallelizing well.”
* Unfortunately, there are no follow-up experiments or discussions on better parallelization. Why is this the case, and how much parallelization improvement does the proposed method bring?

3. Page 5, “let $\mathcal{P}$ represent the set of all permutations of indices” and Eq.(1).
* If $p$ is a permutation, what does $pX^{(i)}$ mean in Eq.(1)? And more importantly, why Eq.(1) defines domain-agnostic? It is not clear as there are no direct citations or proof supporting this claim (the Perceiver paper did not seem to include any specific math statement like this). The reviewer would appreciate more explanation on this part.

---

> ### Author Response · Authors · 2023-11-16
> **Response to Reviewer 36zB (1/3)**
>
> Thank you for your thoughtful feedback. We address your comments below. Please let us know if you have any remaining questions or concerns.
>
> > Originality: the paper did not cite or compare other domain-agnostic SSL methods, either contrastive [3] or masking [4]
>
> Thank you for the additional related work. Regarding comparisons, we have now added [5] as a point of comparison; this is a more recent improvement over [3] mentioned by reviewer 3 on our image domain benchmark settings. We believe [5] is the best prior work in domain agnostic learning, and it far outperforms [3]. We show the results below:
>
> | Architecture                             | Pretraining Method                                  | Domain Agnostic | w/o augmentation | w/ augmentation |
> |-------------------------------------------|------------------------------------------------------|------------------|-------------------|------------------|
> | **Perceiver**                            | Patch                                                | ❌              | **79.6**          | 84.3             |
> | **ResNet**                                | None                                                 | ✔️             | 72.5              | 78.1             |
> | **ResNet**                                | Random Quantization [5] + MoCo-v3                         | ✔️                 | 76.9              | 81.6             |
> | **Perceiver**                            | None                                                 | ✔️             | 52.5              | 62.7             |
> | **Perceiver**                            | Random                                               | ✔️                 | 77.6              | 85.6             |
> | **Perceiver**                            | SMA (ours)                                           | ✔️                 | **78.5**          | **86.1**         |
>
> Despite their method using a ResNet-50 architecture which outperforms our Perceiver Architecture when trained from scratch on ImageNet-100, our method still outperforms this prior art, we add the results from their method to our other ImageNet100 results. Also note that when running their method on our same Perceiver architecture, we found it resulted in embedding collapse, which we theorize is due to the scratch initialization of positional embeddings, and the contrastive loss does not facilitate the learning of these positional embeddings well.  We include a rendering of the updated Imagenet100 table below for quick reference. We apply the method of [5] with the MoCo-v3 framework, as the original paper found MoCo-v3 to work best with their proposed augmentation. Technically the ResNet architecture is not domain-agnostic, but we denote their method as domain-agnostic because their proposed augmentation can be done domain-agnostically.
>
> Regarding [4], we believe this work is more focused on different “domains” within images (real images, sketches, paintings, etc), which is different from our work which focuses on different data modality “domains” (image, text,  For this reason we cannot compare with [4] as their method addresses a different problem.
>
> > Empirical performance improvement can be limited, such as results for the MoleculeNet (Table 2), where the proposed method is sometimes worse than the baseline method (BACE and HIV of Uni-Mol-20M), or improvement can seem limited (Lipo).
>
> Uni-Mol is not a baseline but rather the prior art for this benchmark. Notably, it uses a separate data representation that gives additional information to the model not available to our text-based representation. Compared to the prior art text-based representation we far outperform their best results across all tasks. Additionally, there are many other prior graph-based works that we did not include for the sake of space.

---

> ### Author Response · Authors · 2023-11-16
> **Response to Reviewer 36zB (2/3)**
>
> > While a lower performance is common, since there are only limited baselines (TabNet for the HIGGS benchmark and baselines for the HIGGS benchmark are all from 2021 or prior) and this work is quite empirical, the performance difference can seem noticeable. Nevertheless, the reviewer is not familiar with the benchmarks and did not extensively search for new work with higher results, and will hugely benefit from a response from the authors explaining why the baselines are few.
>
> The prior works for the HIGGS small benchmark span 9 prior works. While it is true that this is less than popular benchmarks such as ImageNet, architecture research for tabular data is less common.
>
> For the HIGGS self-supervised learning benchmark, TabNet is the only prior point of comparison, as it was originally introduced in that paper. Previously we were not aware of any prior works extensively evaluating pre-training, which is why we also compared under this setting so that we would have at least 1 point of external comparison for self-supervised learning. However, we performed a wider literature search and found [6], which explores pretraining methods for tabular data. They use the ROC-AUC metric for the HIGGS subset achieving a best score of 82.1 ROC-AUC. In comparison, our scratch model achieves an ROC-AUC score of 80.78, our random masking model achieves a score of 82.0 ROC-AUC, and our guided masking model achieves a score of 82.8 ROC-AUC. Therefore, our method remains the best in this setting.
>
> Additionally, we found one more comparison point [7], which achieves 73.0 on the HIGGS benchmark, 0.1 higher than the best we previously found FT-T. Previously we also had 5 additional comparison points that we did not include due to them having worse performance than our other points of comparison. The full table with these entries is shown below and in our revised paper.
>
> | Method                           | Accuracy (%) |
> |----------------------------------|--------------|
> | TabNet                       | 71.9         |
> | SNN                           | 72.2         |
> | AutoInt                       | 72.5         |
> | GrowNet                       | 72.2         |
> | MLP                           | 72.3         |
> | DCN2                          | 72.3         |
> | NODE                          | 72.6         |
> | ResNet                        | 72.7         |
> | CatBoost                      | 72.6         |
> | XGBoost                       | 72.7         |
> | FT-T                          | 72.9         |
> | Transformer-PLR [7]             | 73.0         |
> | Transformer Scratch               | 72.7         |
> | Transformer Random Masking        | 74.0         |
> | Transformer Guided Masking (Ours) | **74.8**     |

---

> ### Author Response · Authors · 2023-11-16
> **Response to Reviewer 36zB (3/3)**
>
> > How much overlap did the top attention tokens have? Did the authors quantify them? How does this issue worsen as n increases – a log or linear relationship?
>
> We conducted an experiment in the image domain to empirically evaluate this relationship. We resize an image into different resolutions, thus producing inputs with a different number of tokens, and then compute the overlap ratio. The overlap ratio is computed as (# of tokens in top attentions - # unique tokens in top attentions) / # tokens in top attentions. Therefore, a higher value indicates greater overlap. The graph is shown in the appendix of our updated paper in Figure 4; the overlap ratio increases with the number of tokens and typically ranges between 35% and 50%.
>
> > Page 5, “...while achieving the desired masking ratio and parallelizing well.” Unfortunately, there are no follow-up experiments or discussions on better parallelization. Why is this the case, and how much parallelization improvement does the proposed method bring?
>
> The parallelization is due to performing only a single top-k operation compared to needing a top-k operation for every selected query. Compared to the other additional operations (softmax and a sum along the input length dimension), the top-k operation is the bottleneck in running time on modern GPUs. By requiring only 1 top-k operation we can significantly improve running time.
>
> Empirically, the running time without our proposed approximation on the image domain is 1.13 steps/second. In comparison, when using our approximation, the training runs at 3.58 steps/second, over a 3x increase in throughput. Note that with the approximation, the time to compute masks does not significantly impact train step time (less than 2% of the time per step), but without the approximation, the time to compute masks becomes the bottleneck for large input counts. Additionally, we found no difference in performance between the two methods across domains, and our approximation actually improves stability in our testing.
>
> > And more importantly, why Eq.(1) defines domain-agnostic? It is not clear as there are no direct citations or proof supporting this claim (the Perceiver paper did not seem to include any specific math statement like this). The reviewer would appreciate more explanation on this part.
>
> The Perceiver paper evaluates the domain-agnostic nature of architectures by proposing to shuffle all tokens along with their corresponding positional embeddings. In their experimental results, they denote these results with “shuffled,” where Perceiver is the only architecture that does not experience a performance drop seeing as it is invariant to this form of transformation.
>
> We expand on this definition but define a learning process as domain-agnostic if a consistent shuffling of all tokens results in the same performance following pretraining. One way to understand this definition would be to consider treating any data format as flattened tabular data, given that most tabular methods (with the exception of the ResNet model we compare with) do not use knowledge of how related two columns are. If we were to shuffle all the columns, the performance of the learning algorithm would remain the same. We essentially treat all data in this manner, thereby allowing our method to be seamlessly applied to new domains without any adaptation or knowledge of the domain.
>
> [5] Wu, Huimin, et al. "Randomized Quantization: A Generic Augmentation for Data Agnostic Self-supervised Learning." Proceedings of the IEEE/CVF International Conference on Computer Vision. 2023.
>
> [6] I. Rubachev, A. Alekberov, Y. Gorishniy, and A. Babenko. Revisiting pretraining objectives for tabular deep learning. arXiv preprint arXiv:2207.03208, 2022.
>
> [7] Y. Gorishniy, I. Rubachev, and A. Babenko. On embeddings for numerical features in tabular deep learning. Advances in Neural Information Processing Systems, 35:24991–25004, 2022

---

> ### Author Response · Authors · 2023-11-20
> **Checking In**
>
> We wanted to follow up to see if the response and revisions address your concerns. We would be happy to provide further clarifications and revisions if you have any more questions and if not we would greatly appreciate it if you would reevaluate our paper score. Thank you again for your reviews which helped to improve our paper!

---

> > ### Comment · Reviewer_36zB · 2023-11-22
> > **Response to the authors**
> >
> > The responses include new experimental comparisons and successfully address all of the reviewer's concerns. Therefore, the reviewer updated the rating to 6.

---

### Author Response · Authors · 2023-11-16
**General Response to All Reviewers and Summary of Revisions**

We thank all the reviewers for their thoughtful feedback, which we believe has helped to further improve the paper.

At the request of the reviewers, we have run new experiments and comparisons. These include:
* Comparisons with [1], the prior state-of-the-art in domain-agnostic self-supervised learning (36zB, ijD8).
* Comparisons against [2, 3] in the HIGGS benchmark (36zB).
* Information about masking ratio sweeps (zapB, ijD8)
* Tabular results on audio features year prediction [4] (Md4o)
* Per-step running time to demonstrate the computational efficiency of our method (36zB, Md4o, eSva)
* Experiment demonstrating how the amount of overlap relates to the number of inputs in the image domain (36zB)

In addition, we have added a new figure detailing the end-to-end structure of our entire method to make our paper more self-contained while improving clarity in the appendix (eSva).

We address specific concerns in individual replies to each reviewer. If there are any remaining questions or concerns, please let us know!

[1] Wu, Huimin, et al. "Randomized Quantization: A Generic Augmentation for Data Agnostic Self-supervised Learning." Proceedings of the IEEE/CVF International Conference on Computer Vision. 2023.

[2] I. Rubachev, A. Alekberov, Y. Gorishniy, and A. Babenko. Revisiting pretraining objectives for tabular deep learning. arXiv preprint arXiv:2207.03208, 2022.

[3] Y. Gorishniy, I. Rubachev, and A. Babenko. On embeddings for numerical features in tabular deep learning. Advances in Neural Information Processing Systems, 35:24991–25004, 2022

[4] B. McFee, T. Bertin-Mahieux, D. P. Ellis, and G. R. Lanckriet. The million song dataset challenge. In Proceedings of the 21st International Conference on World Wide Web, pages 909–916, 2012.

---

### Meta-Review · Area_Chair_hUfB · 2023-12-12

**Metareview:**

The rebuttal addressed most of the concerns, so now all reviewers favor accepting the paper. hence, it will be accepted.

**Justification For Why Not Higher Score:**

After rebuttal, all reviewers rated it at level 6, so I believe poster is the right decision. The main weaknesses are: simple experiments, lack of sufficient motivation, and missing related work.

**Justification For Why Not Lower Score:**

I agree with the reviewers that the work is solid and can be accepted in ICLR.

---

### Decision · Program_Chairs · 2024-01-16

Accept (poster)